ecology, evolution

detoxification, human intervention, metal pollution, microbial communities, selection, siderophores

**Author for correspondence:**
Elze Hesse
e-mail: elzehesse@gmail.com

†Present address: Unité Géomicrobiologie et Monitoring Environnemental, D3E, BRGM (French Geological Survey), 3 av. Claude-Guillemin—BP 36009, 45060 Orléans Cedex 2, France.

# Anthropogenic remediation of heavy metals selects against natural microbial remediation

Elze Hesse[1], Daniel Padfield[1], Florian Bayer[1], Eleanor M. van Veen[2], Christopher G. Bryan[2,†] and Angus Buckling[1]

[1]ESI and CEC, Biosciences, University of Exeter, and [2]Camborne School of Mines, CEMPS, University of Exeter, Penryn Campus, Cornwall TR10 9FE, UK

(iD) EH, 0000-0002-1900-7136

In an era of unprecedented environmental change, there have been increasing ecological and global public health concerns associated with exposure to anthropogenic pollutants. While there is a pressing need to remediate polluted ecosystems, human intervention might unwittingly oppose selection for natural detoxification, which is primarily carried out by microbes. We test this possibility in the context of a ubiquitous chemical remediation strategy aimed at targeting metal pollution: the addition of lime-containing materials. Here, we show that raising pH by liming decreased the availability of toxic metals in acidic mine-degraded soils, but as a consequence selected against microbial taxa that naturally remediate soil through the production of metal-binding siderophores. Our results therefore highlight the crucial need to consider the eco-evolutionary consequences of human environmental strategies on microbial ecosystem services and other traits.

## 1. Introduction

Intervention strategies aimed at improving human health, agriculture, biotechnology and the environment inevitably impact microbial communities, even in situations where microorganisms are not the direct targets of intervention. Given the potential of microbial populations and communities to respond rapidly to any environmental change [1–5], it is crucial to consider both the short- and long-term effects resulting from ecological species sorting and adaptation. Of particular concern is if longer-term responses reduce the efficacy of intervention or result in negative consequences. This might occur if intervention opposes the naturally human-beneficial characteristics of microbes. This study investigates such a scenario in the context of raising pH using lime-containing materials, a common intervention practice aimed at reducing heavy metal toxicity [6].

Heavy metals (metals and metalloids with a density above $5 \, \mathrm{g}^{-1} \, \mathrm{cm}^3$) are ubiquitous components of the Earth's crust [7]. As a result of soaring demands for minerals [8], large parts of the world are currently mined for valuable mineral deposits, leaving a legacy of untreated mining waste [9]. In addition, agricultural practices such as application of sewage sludge and phosphate fertilizers have led to increased heavy metal concentrations in the environment [10,11]. Heavy metals typically persist for a long time after their introduction [12] and can adversely affect human, plant and wildlife at high concentrations [13]. Consequently, there is increasing interest in remediating metal-contaminated environments. Acidic conditions often prevail in mine-degraded sites, and the bioavailability of many heavy metals is typically increased under such conditions [6]. Hence, lime-containing materials are commonly applied to severely metal-contaminated soils to neutralize soil pH [14], immobilize heavy metals [15] and thereby facilitate natural regeneration [6,16,17].

Microbial communities inhabiting contaminated soils have evolved various resistance mechanisms, including sequestration, efflux and extracellular chelation [18–21]. Crucially, some of these mechanisms—notably chelation—also act to remediate the environment. Much of this chelation is carried out by siderophores—low-molecular-weight high-affinity iron chelators that are produced and secreted by many microorganisms in response to iron deprivation [22]. While the canonical function of siderophores is iron scavenging, these compounds also bind to other metals, thereby preventing their uptake into cells and rendering the environment less toxic [23,24]. Our recent work has shown that siderophore-producing microbial taxa are selectively favoured in metal-polluted soils [25]. This is despite opportunities for non-producing 'cheats' to be favoured, given that siderophores provide both within- and across-species protective benefits against toxic metals [26]. It therefore follows that raising pH may select against this natural decontamination process, resulting in lower average siderophore production across the community. However, iron—a major factor limiting growth—becomes increasingly insoluble in basic environments ($pH > 6.5$) [27]. Hence, liming might selectively favour microbial taxa that produce siderophores as a result of iron deprivation. This in turn might select for intra-specific [28], or even in some cases inter-specific [29], exploitation of iron-bound siderophores. The net effect of liming on siderophore production (and the associated natural decontamination) is therefore unclear.

We experimentally determine how liming of acidic mine-degraded soils influences microbial community function (i.e. production of metal-chelating siderophores) and composition. We collected 30 distinct soil samples in a historical mining area to determine whether liming has a consistent effect across soils varying widely in their initial pH, metal content and community composition [25]. Using a paired design, we subjected each soil to two different selection regimes by incubating soil microcosms for 12 weeks with and without hydrated lime. Soil characteristics and siderophore production were quantified before and after experimental manipulation. In addition, we measured changes in community composition by sequencing the 16S rRNA gene (which covers spore-forming bacteria like *Bacillus*) [30]. Our results demonstrate that liming opposes selection on natural decontamination traits by favouring microbial taxa that produce few or no siderophores, leading to a net decrease in community siderophore production.

## 2. Material and methods

### (a) Study site and soil sampling

We collected 30 soil samples along a natural metal gradient located in a disused poly-metallic mine in the Poldice Valley, Cornwall, UK (N: 50°14.10; W: −5°10.23). Soils were collected and processed as described previously [25], after which soil acidity was quantified on the same day (see below). A small fraction of soil per sample was stored at −80°C for phenotypic assays and DNA extractions; the remainder was used to set up a selection experiment (see below).

### (b) Selection experiment

To test whether liming selects against siderophore production in natural microbial communities, we set up experimental microcosms by placing 30 g of soil per sample in duplicate 90 mm Petri dishes. Using a paired design, we imposed two different selection regimes: a single dose of hydrated lime (100 mg of Verve Garden lime dissolved in 5 ml of sterile ddH₂O) was added to half of the paired microcosms and 5 ml of sterile $dd$H₂O to the remainder. Lime-treated and control microcosms ($n = 60$) were incubated in an environmental chamber at 26°C and 75% relative humidity and were kept moist throughout. After 12 weeks of incubation, we collected samples to (i) quantify soil acidity and heavy metal concentrations, (ii) characterize microbial communities and (iii) prepare freezer stocks for siderophore assays. Freezer stocks were prepared by vortexing 1 g of soil for 1 min with 6 ml of M9 buffer and sterile glass beads, after which the soil washes were stored at −80°C in a final concentration of 25% glycerol.

### (c) Soil characterization

Soil acidity was quantified before and after experimental manipulation by suspending 1 g of soil per sample in 5 ml of 0.01 M CaCl₂, which was then shaken for 30 min and left to stand for 1 h, after which pH was measured using a Jenway 3510 pH meter (Stone, UK) [31]. For experimental soils, we also quantified soluble metal concentrations using the detachment procedure described previously [32,33]. Briefly, we suspended 5 g of soil per microcosm in 5 ml of ddH₂O in 50 ml falcon tubes that were gently shaken to disperse soil aggregates and centrifuged for 1 min at 300 r.p.m. to remove solids. 1 ml of supernatant was transferred to Eppendorf tubes and re-spun at 3000 r.p.m. for 3 min to remove final solids. The resulting supernatants were 1 : 1 diluted in 1% HCl, after which solution chemistry (Ag, Al, As, Cd, Co, Cu, Cr, Fe, Ga, Mg, Mn, Ni, Pb, Sn, Ti and Zn) was determined using ICP-MS. As the overwhelming majority of soil microbes reside within interstitial spaces in pore networks [34,35], the presence of soluble metals in pore water is a good proxy of metal availability and hence toxicity [36].

### (d) Microbial community characterization

To determine how community composition varied across soils, we extracted genomic DNA from 250 mg soil per sample (all stored in buffer and C1 solution at −80°C) using the MoBio Powerlyzer PowerSoil DNA isolation kit (Carlsbad, CA, USA), following the manufacturer's protocol with the bead beating parameter set to 4500 r.p.m. for 45 s. Samples were additionally cleaned using the Zymo OneStep PCR Inhibitor Removal Kit following the manufacturer's protocol. The integrity of DNA was confirmed using 1% TAE agarose gels stained with $1 \times$ Redsafe DNA Stain (20 000 ×), yielding a total of 78 high quality DNA samples (i.e. samples 2, 8, 11 and 15 were excluded as DNA yield was not of sufficiently high quality for amplicon sequencing).

Sequencing of amplicons of the V4 region of the 16S rRNA gene using the Illumina MiSeq 16S Ribosomal RNA Gene Amplicons Workflow was undertaken by the Centre for Genomic Research (Liverpool, UK) using the following primers [37]:

Forward: 5′ACACTCTTTCCCTACACGACGCTCTTCCGATCTN NNNNGTGCCAGCMGCCGCGGTAA3′
Reverse: 5′GTGACTGGAGTTCAGACGTGTGCTCTTCCGATCT GGACTACHVGGGTWTCTAAT3′.

Briefly, 5 µl of DNA (mean ± s.d. concentration = 15.99 ± 11.80 ng µl⁻¹) entered a first round of PCR with cycle conditions 20 s at 95°C, 15 s at 65°C, 30 s at 70°C for 10 cycles, followed by a final 5-min extension at 72°C. The primer design incorporates a recognition sequence to allow a secondary nested PCR step. Samples were first purified with Axygen SPRI beads before entering the second PCR performed to incorporate Illumina sequencing adapter sequences containing indexes (i5 and i7) for sample identification. A second round of PCR was performed using the same conditions as above for a total of 25 cycles. Samples were purified using Axygen SPRI beads before being

quantified using Qubit and assessed using a fragment analyser. Successfully generated amplicon libraries were taken forward. Final libraries were pooled in equimolar amounts using Qubit and fragment analyser data, and size selected on the Pippin prep using a size range of 300–600 bp. The quantity and quality of each pool was assessed by Bioanalyzer and subsequently by qPCR using the Illumina Library Quantification Kit from Kapa on a Roche Light Cycler LC480II according to manufacturer's instructions. The template DNA was denatured according to the protocol described in the Illumina cBot User guide and loaded at 8.5 pM concentration. To help balance the complexity of the amplicon library 15% PhiX was spiked in. The sequencing was carried out on one lane of an Illumina MiSeq at $2 \times 250$ bp paired-end sequencing with v2 chemistry.

The raw Fastq files were trimmed for the presence of Illumina adapter sequences using CUTADAPT version 1.2.1 [38], using the option $-$O 3 (i.e. 3′ end of any reads matching the adapter sequence for 3 bp or more were trimmed). Reads were further trimmed using SICKLE version 1.200 with a minimum window quality score of 20. Reads shorter than 20 bp after trimming were removed. If only one of a read pair passed this filter, it was included in the R0 file. We then processed and analysed the trimmed sequence data in R using the packages 'dada2' and 'phyloseq' [39,40]. Following the standard full stack workflow [40], we estimated error rates, inferred and merged sequences, constructed a sequence table, removed chimeric sequences and assigned taxonomy. During processing, forward and reverse reads were truncated between 25–250 and 25–230 nucleotide positions, respectively, due to poor quality scores. Assembled amplicon sequence variants (ASVs) were assigned taxonomy using the Ribosomal Database Project [41].

We estimated the phylogenetic tree using *FastTree* which uses approximate maximum likelihood to estimate phylogeny from nucleotide alignments [42]. We then further quality controlled processed sequencing data before analyses were undertaken. We filtered out all reads that had not been assigned to the phylum level, any ASVs that were present in less than 5% of all samples and any reads that were assigned as either cyanobacterial or chloroplast origin. Processing and filtering steps resulted in all the 78 samples remaining for downstream analysis, with a maximum number of reads in a sample of 650 204, minimum of 21 594 and mean of 61 609. Amplicon sequencing data have been deposited as ENA Project PRJEB28850 (https://www.ebi.ac.uk/ena/data/view/PRJEB28850).

## (e) Siderophore assays

For each unique soil–treatment combination ($n = 90$), we quantified siderophore production by plating out serial-diluted freezer stocks on LB agar plates supplemented with Nystatin ($20 \, \mu g \, ml^{-1}$ final concentration) to suppress fungal growth. Plates were incubated at 28°C for 48 h, after which 24 individual isolates per sample were randomly selected and grown independently in 2 ml of iron-limited CAA medium (5 g Casamino acids, 1.18 g $K_2HPO_4.3H_2O$, 0.25 g $MgSO_4.7H_2O \, l^{-1}$, supplemented with 20 mM $NaHCO_3$ and $100 \, \mu g \, ml^{-1}$ human apotransferrin) [43]. After 48 h of growth at 28°C, we spun down cultures for 15 min at 3000 r.p.m. and assayed supernatants to determine the extent of iron chelation using the liquid CAS assay described by Schwyn & Neilands [44], modified such that one volume of $ddH_2O$ was added to the CAS assay solution [45]. Siderophore production per isolate was estimated using $[1 - (A_i/A_{ref})]/(OD_i)$, where $OD_i$ = optical density at 600 nm and $A_i$ = absorbance at 630 nm of the assay mixture (supernatant + CAS solution) and $A_{ref}$ = absorbance at 630 nm of reference uncultured medium mixture (CAA + CAS solution). We measured siderophore production under common garden conditions to avoid confounding effects of environmental variation *in situ*, causing both differential siderophore induction and soil metal-chelating activities.

## (f) Statistical analyses

The effect of liming on soil acidity and heavy metal content (non-ferrous and total soluble metals) was tested using linear mixed effects models ('lmer' function from the 'lme4' package) [46], with random intercepts fitted for individual samples ($n = 30$) to account for soil-specific dependencies. We used a similar approach to test for the effect of liming on siderophore production. In general, full models were simplified by sequentially eliminating non-significant terms ($p > 0.05$) following a stepwise deletion procedure, after which the significance of the explanatory variables was established using likelihood ratio tests, which were $\chi^2$ distributed. In the case of significant treatment effects, Tukey contrasts were computed using the 'glht' function from the R package 'multcomp' [47], with $\alpha < 0.05$. The concentration of soluble metals varied widely, ranging from 0 to $1042 \, mg \, l^{-1}$ of pore water (figure 2). To test how liming affected the availability of rare metals, in particular, we calculated standardized differences between paired samples by dividing sample-specific metal quantities by the overall mean of each metal. We then used one-tailed *t*-tests to compare standardized treatment differences to zero, corrected for multiple testing ('p.adjust' with method = 'fdr'). We removed some metals (As, Ga, Pb, Sn and Ti) from these analyses as the majority of samples contained undetectable levels (electronic supplementary material, table S1).

To compare the composition of the microbial communities, we examined the impact of community origin and treatment (i.e. ancestral sample, liming or control) on the weighted Unifrac distance, which weights the branches of the phylogenetic tree based on the abundance of each ASV. Differences in composition between communities were analysed using the R packages 'phyloseq' and 'vegan' [48]. Permutational ANOVA tests were run using the 'adonis' function in the 'vegan' package using community origin and treatment as main effects and weighted Unifrac distance as a response term with 9999 permutations. We controlled for the nestedness of the data (i.e. treatment within community origin) by limiting the shuffling of each permutation to within samples of the same community origin only. We then did pairwise permutational ANOVAs to better understand which treatments were different from each other by running the same approach with only treatment as a main effect and filtering each treatment out in turn. Significance of *p*-values was then determined using Bonferroni correction, with $\alpha < 0.05$. We used R v. 3.1.3 for all analyses (R Development Core Team; http://www.r-project.org). Raw phenotypic data are available from the Dryad Digital Repository: https://doi.org/10.5061/dryad.843814d [49].

## 3. Results

## (a) Liming reduces soil acidity and total non-iron metal availability

Soils across the metal gradient varied widely in their pH (lmer: estimated standard deviation of community origin random effect = 0.97), being predominantly acidic before experimental manipulation (figure 1a). Liming had the desired effect of raising pH ($\chi^2_2 = 64.85$, $p < 0.001$; figure 1b), such that pH was significantly greater in lime-treated soils (L; $\overline{pH} = 6.30$, 95% CI = 5.98–6.62) compared with ancestral ($T_0$; $\overline{pH} = 5.38$, 95% CI = 4.98–5.79) and control (C; $\overline{pH} = 5.43$, 95% CI = 5.00–5.86) soils (Tukey contrasts for C $-$ $T_0$: $z = 0.53$ and $p = 0.86$, L $-$ $T_0$: $z = 9.46$ and $p < 0.001$ and L $-$ C: $z = 8.93$ and $p < 0.001$).

Iron was by far the most common metal across all samples and liming did not restrict its availability (paired *t*-test: $t = 0.15$, d.f. = 29, $p = 0.88$; mean [95% CI] iron availability for C = 77.17 [31.92, 122.41] and L = 81.04 [50.88, 111.20] $mg^{-1}$ l pore water). As a consequence, total metal availability did not differ between treatments ($\chi^2_1 = 0.96$, $p = 0.33$; figure 2b).

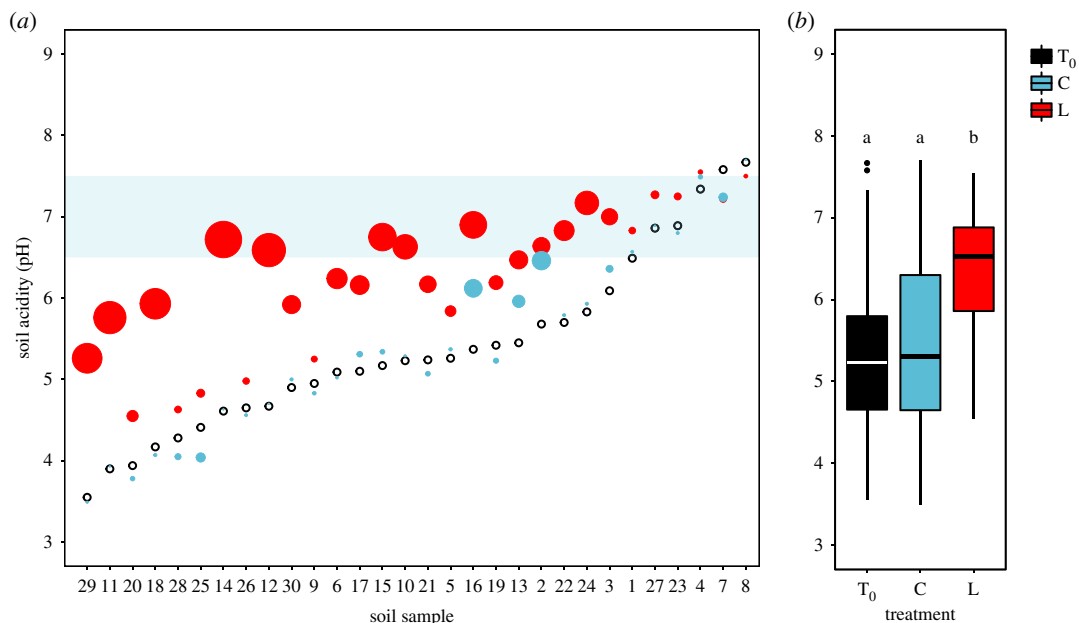

**Figure 1.** Variation in soil acidity (pH) in a historical mining site. (a) Soils were predominantly acidic (T₀, black symbols), falling well below the blue shaded area of pH neutrality. Symbol size indicates the extent to which the lime (L, red symbols) and control (C, light blue symbols) treatments altered initial soil acidity (T₀). Note that the effect of lime addition is greater for more strongly acidic soils. (b) Box whisker plot showing a reduction in soil acidity in response to liming. Boxes depict the upper and lower quartiles of the treatment-specific raw data with the centre line showing the median and whiskers providing a measure of the 1.5× interquartile range. Letters denote significant Tukey contrasts, with $\alpha < 0.05$. (Online version in colour.)

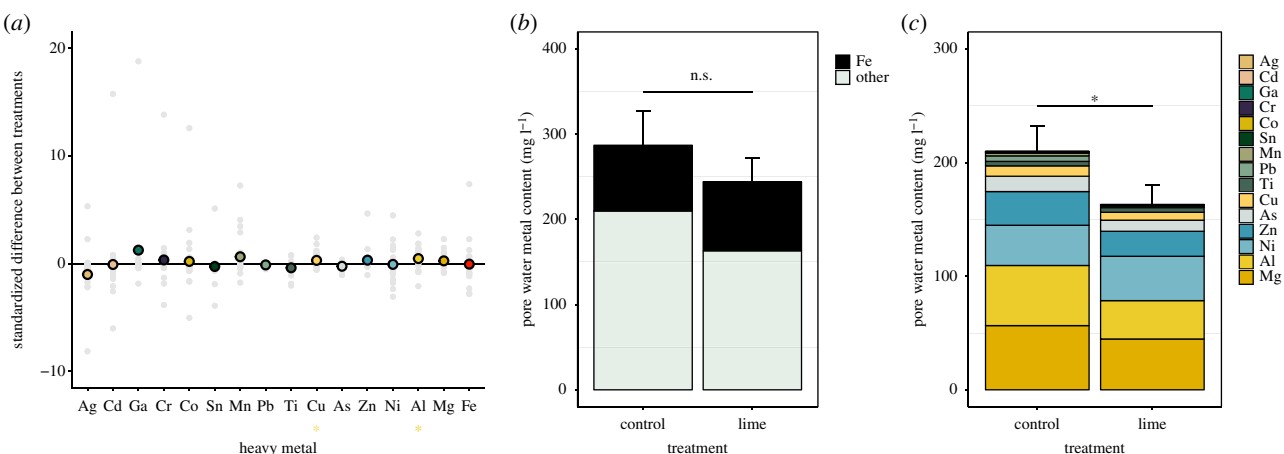

**Figure 2.** Liming reduces the availability of a subset of non-iron metals. (a) Plot depicting mean (coloured symbols) and individual (grey symbols) standardized differences between paired soil microcosms, where positive values indicate a reduction in dissolved metal concentrations in limed versus control soils. Stars summarize the significance of this effect, estimated using one-tailed $t$-tests corrected for multiple testing (*$p < 0.05$). Bar plots demonstrating (b) a non-significant effect of liming on total metal availability (mean ± 1 s.e.), partitioned into iron (black) and non-iron (mint green) metals, and (c) a significant reduction in total non-iron metal availability (mean ± 1 s.e.) in limed soils, partitioned into individual metals, which are arranged based on their mean across-treatment availability, with magnesium (Mg) being most common. Asterisk denotes significant Tukey contrast, with $\alpha < 0.05$. (Online version in colour.)

However, the total availability of non-ferrous metals was significantly lower in lime-treated soils ($\chi^2_1 = 3.96$, $p = 0.047$; figure 2c). As a likely result of differing metal solubilities and co-precipitation [12], the effect of liming varied greatly across metals (figure 2a and electronic supplementary material, table S1). Notably, while none of the quantified metals was more readily available in lime-treated soils, liming did significantly reduce the level of soluble copper (Cu) and aluminium (Al), both of which can be toxic at high concentrations [50].

### (b) Liming changes microbial community composition
The majority of the variation in community composition between samples (i.e. weighted Unifrac distance) was accounted for by the natural metal gradient (PERMANOVA,

$F_{25,50} = 14.45$, partial $R^2 = 0.85$, $p < 0.001$; figure 3). While these large between-community differences masked the effect of liming to a certain extent, the relative abundance of several common phyla—including members of the Acidobacteria, Chloroflexi and Gemmatimonadetes—did change in response to liming (electronic supplementary material, figure S1). In agreement, our analysis confirms that different taxa were favoured across different treatments (PERMANOVA, $F_{2,50} = 6.20$, partial $R^2 = 0.03$, $p < 0.001$): when decomposed into multiple pairwise comparisons, community composition differed significantly between liming and control (Bonferroni-corrected $p = 0.02$), and liming and ancestral community (Bonferroni-corrected $p < 0.001$), but not between control and ancestral community (Bonferroni-corrected $p = 0.41$).

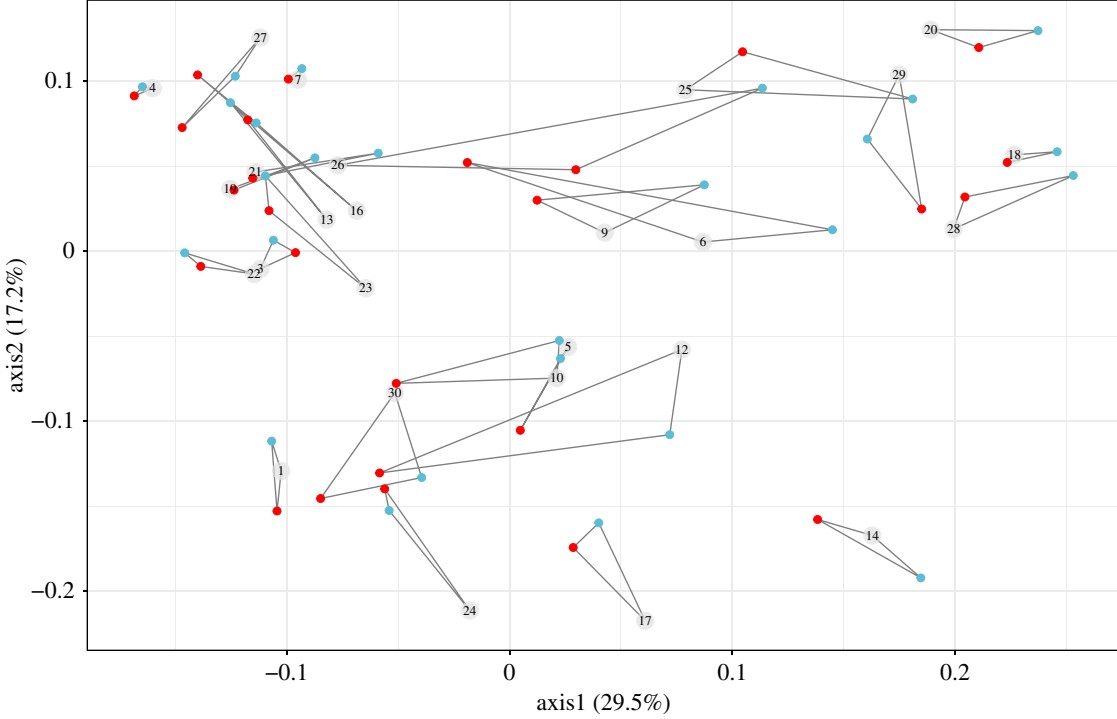

**Figure 3.** Principal coordinate analysis (PCoA) plot based on weighted Unifrac distance of microbial ASVs between communities. The percentage of variation explained is shown on each axis, calculated from the relevant eigenvalues. Communities with the same community origin are joined with straight lines. Although community origin drives most of the variation in community composition, the addition of lime (red circles) also significantly changes community composition compared to the ancestral (labelled in numbered grey circle) and control (light blue circles) communities. (Online version in colour.)

## (c) Liming selects against detoxifying siderophores

Microbial communities naturally varied in their siderophore production (lmer: random intercept variance of microbial community = 0.15; figure 4a), with all ancestral communities containing multiple siderophore-producing isolates (figure 4b). Crucially, liming strongly selected against siderophore production ($\chi^2_2 = 2247.6$, $p < 0.001$; figure 4c). Mean siderophore production was significantly lowered in lime-treated (mean = −0.25, 95% CI = −0.32 to −0.18) compared to ancestral (mean = 0.31, 95% CI = 0.24–0.39) or control soils (mean = 0.56, 95% CI = 0.48–0.64) (Tukey contrasts for C–$T_0$: $z = 18.85$, L–$T_0$: $z = -42.84$ and L–C: $z = -61.60$, all $p < 0.001$). This pattern was driven by lime selectively favouring non-siderophore-producing isolates (figure 4b,c). The increase in siderophore production in control soils through time (figure 4c) could be driven by a range of abiotic and biotic changes associated with growth under laboratory conditions.

## 4. Discussion

Our findings demonstrate liming consistently selected against high siderophore-producing taxa across a range of metalliferous soils despite the fact that microbes can produce multiple siderophores with very different metal affinities [22,24], and ancestral communities varied widely in their initial composition and pH. Our results suggest that this effect is driven primarily because liming reduced the availability of toxic metals, and hence there is little benefit of siderophore-mediated detoxification. An additional explanation is that liming, by reducing metal availability, increased the need for siderophore-mediated iron acquisition, and there was selection for non-producing cheats exploiting iron-bound siderophores produced by other community members. However, this is unlikely to be important,

because most siderophore–iron complexes can be taken up by few taxa in addition to the producer [51] (unlike decontamination, which does not involve uptake) and, more importantly, unlike most metals, liming did not decrease the availability of iron. In other words, selection imposed on siderophore production as a consequence of iron availability would have differed little between limed and control soils. We also cannot rule out the possibility that siderophore production is not under direct selection, but is purely a correlated response to the well-documented [52,53] compositional changes resulting from changes in pH. However, given that siderophores are known to detoxify [23], and there are both intra- and inter-species changes in siderophore production in response to metal addition in the absence of large pH changes [25], this would seem highly unlikely. Intervention practices that buffer the effects of metal toxicity are therefore likely to select against other microbial resistance traits that remediate the environment (e.g. metal sequestration), irrespective of whether these traits primarily benefit the actor or confer cooperative resistance to other community members [54].

Our previous work has shown that changes in community-wide siderophore production are primarily shaped by species sorting [25], although rapid evolutionary change—both via mutation and horizontal gene transfer [55]—cannot be ruled out. While liming did have a significant effect on community composition, the impact of liming was relatively small compared with that resulting from historical environmental conditions, and liming often selected for different taxa between communities. This would imply that lime addition simply selects against high siderophore-producing microbial taxa, because the marginal costs of siderophore production [56] become greater when not needed for remediation.

Our current and previous findings demonstrate that population turnover in soil microbial communities is high [57,58]

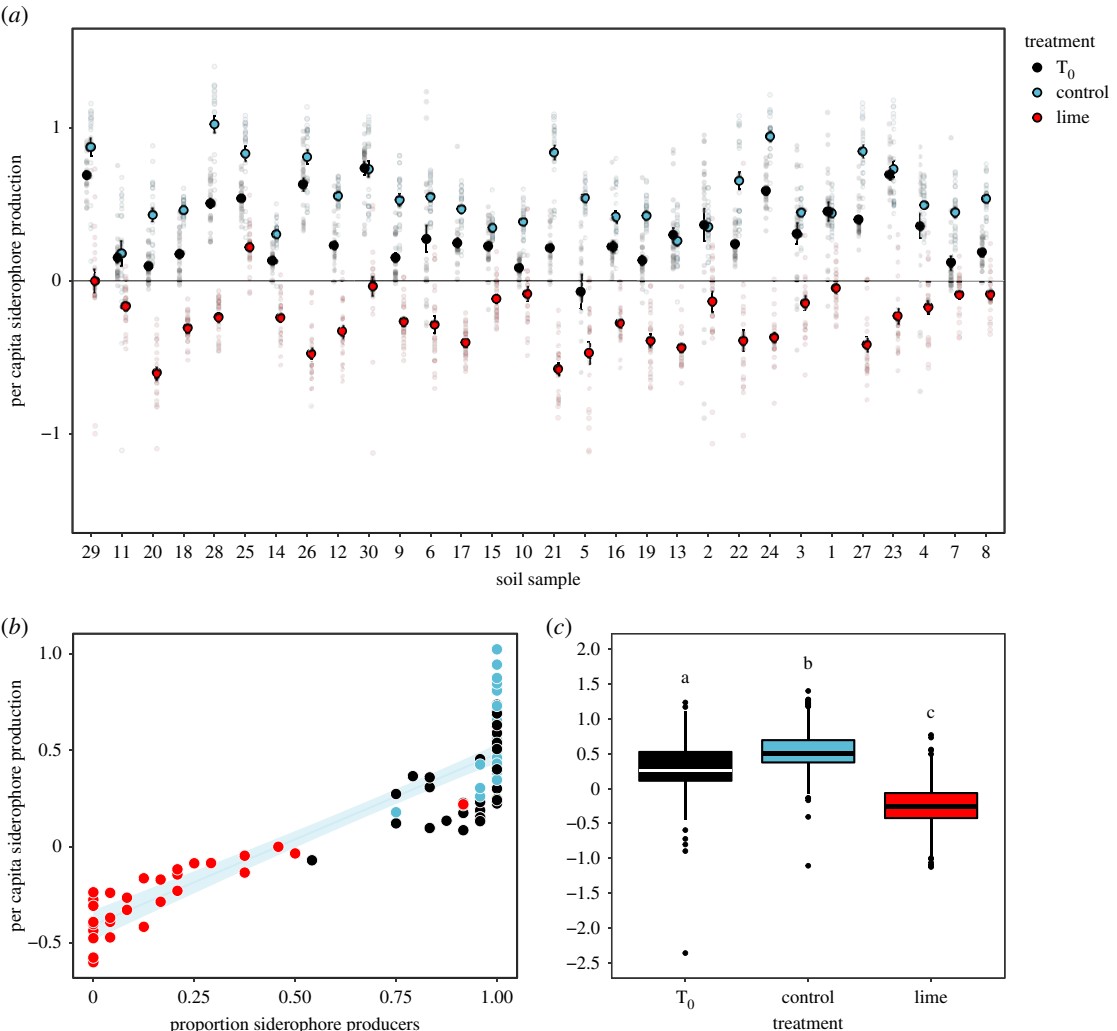

**Figure 4.** Liming selects against microbial siderophore production. (a) Variation in siderophore production across thirty microbial communities as a function of selection regime, where small dots depict raw data and large coloured dots provide a measure of mean per capita siderophore production $\pm$ 1 s.e. in ancestral (black), limed (red) and control (light blue) soils, ranked based on their initial soil acidity. (b) Plot demonstrating a positive linear relationship between mean siderophore production and the proportion of producers within each community ($n = 90$). Line and shaded area depict fitted relationship [$\pm 1$ s.e.] pooled across treatments: $y = -0.41\ [0.04] + 0.89\ [0.05]\ x$; $F_{1,88} = 337.8$, $p < 0.001$; adjusted $R^2 = 0.79$. Coloured symbols denote the different selection regimes, where black, red and light blue are ancestral, limed and control soils, respectively. (c) Boxplot summarizing treatment-specific variation in siderophore production, where different letters denote significant Tukey contrasts, with $\alpha < 0.05$. (Online version in colour.)

and that siderophore production can change rapidly through time [25], which is perhaps unsurprising as it is largely driven by species sorting. As a consequence, ceasing lime addition will be likely to result in restoration of microbial siderophore production. This raises the question as to whether there are any negative consequences associated with chemical remediation on the medium to long-term. While liming reduces soil acidity and metal solubility, it does not actually remove toxic metals from the environment. Siderophores or siderophore-producing microbes, on the other hand, can help with heavy metal removal when combined with the use of hyper-accumulating plants (phytoextraction) [59–62]. Specifically, metal uptake by these plants is typically enhanced if metals are bound to microbial siderophores [63–68]. While liming of metalliferous soils has been shown to increase plant yield in agricultural settings [69], it does indeed appear to reduce plant metal uptake, with both correlational and experimental studies suggesting lower trace metal accumulation at high (greater than 7) compared with low (less than 7) soil pH [62,70]. Whether this pH-mediated metal uptake is caused by changes in siderophore production needs to be

addressed experimentally; lime-mediated selection against detoxifying siderophores could negate the benefits arising from enhanced biomass accumulation, thereby hampering phytoextraction efficacy.

To conclude, anthropogenic pollution is a major problem worldwide. While there is a pressing need to remediate polluted ecosystems, our findings indicate that liming—a common intervention practice—opposes selection operating on natural microbial detoxification. Microbes facilitate many of the processes mediating ecosystem services, including decomposition and mineralization, disease causation and suppression, and pollutant removal [4]. Understanding the eco-evolutionary consequences of human intervention on microbial traits (e.g. detoxification, resistance) is key for the engineering of evolutionary resilient microbial communities, having important implications for phytoremediation, with further relevance to global human health and industry [56].

Data accessibility. Data available from the Dryad Digital Repository: https://doi.org/10.5061/dryad.843814d [49].
Authors' contributions. E.H., E.M.v.V. and A.B. conceived and designed the experiment. E.H. collected the data. E.H., F.B. and D.P. carried

out the data analyses. C.G.B. provided new perspectives. E.H. and A.B. wrote the first draft of the manuscript, and all authors contributed substantially to revision.
Competing interests. We declare we have no competing interests.

Funding. A.B. acknowledges support from the Royal Society. This work was funded by the AXA Research Fund, BBSRC and NERC.
Acknowledgements. We thank Sharon Uren for running samples on ICP-MS and Jesica Soria Pascual for helping out with siderophore assays.

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
