## [Reviewer comments · Proceedings of the Royal Society B: Biological Sciences]

Review History

RSPB-2019-0804.R0 (Original submission)

Review form: Reviewer 1

Recommendation

Accept as is

Scientific importance: Is the manuscript an original and important contribution to its field?

Excellent

General interest: Is the paper of sufficient general interest?

Good

Quality of the paper: Is the overall quality of the paper suitable?

Excellent

Is the length of the paper justified?

Yes

Should the paper be seen by a specialist statistical reviewer?

No

Do you have any concerns about statistical analyses in this paper? If so, please specify them explicitly in your report.

No

It is a condition of publication that authors make their supporting data, code and materials available - either as supplementary material or hosted in an external repository. Please rate, if applicable, the supporting data on the following criteria.

Is it accessible?

Yes

Is it clear?

Yes

Is it adequate?

Yes

Do you have any ethical concerns with this paper?

No

Comments to the Author

In this respot, the authors performed a simple microcosm experiment to assess the effect of anthropogenic disturbance (liming) on a number of microbial communities' ability to bioremediate pollution (heavy metal through siderophores). I really must commend the authors on an extremely well reasoned, designed and explained study. There is nothing I would amend to this study hence the recommendation to accept as is.

Review form: Reviewer 2

Recommendation

Reject - article is scientifically unsound

Scientific importance: Is the manuscript an original and important contribution to its field?

Good

General interest: Is the paper of sufficient general interest?

Acceptable

Quality of the paper: Is the overall quality of the paper suitable?

Good

Is the length of the paper justified?

Yes

Should the paper be seen by a specialist statistical reviewer?

Yes

Do you have any concerns about statistical analyses in this paper? If so, please specify them explicitly in your report.

No

It is a condition of publication that authors make their supporting data, code and materials available - either as supplementary material or hosted in an external repository. Please rate, if applicable, the supporting data on the following criteria.

Is it accessible?

N/A

Is it clear?

N/A

Is it adequate?

N/A

Do you have any ethical concerns with this paper?

No

Comments to the Author

Manuscript: Anthropogenic remediation of heavy metals selects against natural microbial remediation by Hesse et al.

Dear authors,

Thank you for your manuscript "Anthropogenic remediation of heavy metals selects against natural microbial remediation". Here you study an interesting question whether artificial heavy metal detoxification strategies selects against microbial communities which perform natural remediation. The work has a great scope in various areas including microbial ecology, evolution and, health.

The observations from the study are: (a) liming increases pH of the soil (makes it slightly alkaline); (b) it decreases availability of metals (although only in 2 out of 11 metals studied); (c) iron was most abundant and importantly upon liming it's availability is not decreased; (d) after 12 weeks of liming treatment the microbial communities changed from mainly siderophore producers to non-producers. From these, authors conclude that anthropogenic strategies such as liming selects against natural detoxification.

The experimental design, sample size and the statistical analysis used by the authors are appropriate.

Here are my concerns:

(a) First of all, liming itself does not appear to be a great strategy for detoxification as it decreases the availability of only a few metals. This, however, is not the authors claim but I wonder if it would have been appropriate to study the effect of anthropogenic strategies which have shown to be efficient in detoxification.

(b) There is a correlation between increased pH (upon liming) and selection for siderophore non-producers. Siderophores are produced for chelating iron. Assuming siderophores also chelate other metals (to a large extent?) the mere selection of siderophore non-producers does support the conclusion that anthropogenic strategies selects against natural detoxification. Moreover, authors indirectly claim that natural remediation is efficient and artificial strategies affect this process. I guess the artificial strategies are used because natural process is either very slow or probably not efficient to detoxify the huge heavy metal load coming out from mining, etc.

Other concerns (as they appear in text):

1. Line 99: "To test whether liming selects against detoxification in natural microbial communities". I think this should be phrased as "To test whether liming selects against siderophore non-producer..." or something else like this. This is because detoxification is not under selection. It (if at all) is a consequence of selection on a particular bacterial 'phenotypes'.
2. Line 109: "after which the supernatant was stored..." Please mention if supernatant was treated in some way (filtered/centrifuged). This is because once vortexed the soil mixture probably was highly turbid and some floating particles.
3. Line 149: "Briefly, 5 μ l of DNA.." mention DNA concentration or total DNA (either ug or ug/ul).
4. Line 200: "mixture (CAA+CAS)." I wonder if this should have been CAA + water as the authors diluted assay solution with water.
5. Line 238: "(a) Liming reduces soil acidity and non-iron metal availability." Although in a relatively small number of samples but how do we explain increased pH in CONTROLS? Could this be because of natural properties of the bacterial communities in those samples? How do we explain the variation in red symbols? Here, the size of the symbol indicates change in pH which I thought should have been uniform. Besides, in some cases after liming the pH is decreased.
6. Line 246: "Iron was by far the most common metal across all samples and liming did not restrict its availability". Based on this result I wonder if the major conclusion of the MS can be interpreted differently. Authors claim that liming selects against natural detoxification. First of all, this should be liming selects siderophore non-producers. But, importantly, if iron availability is not affected and as authors point out the iron was abundant in their soil samples then one would expect siderophore non-producers to be favoured (probably because of the cost of basal level siderophore production).
7. Line 250: However, the availability of non-ferrous metals was significantly lower in lime-treated soils (..... $P = 0.047$). Authors probably will agree that the availability of non-ferrous metal appears to be statistically significant ($p = 0.047$) however these differences appear to be minor and probably does not mean much from the biology perspectives.
8. Line 253: "Notably, while none of the quantified metals was more readily available in lime-treated soils, liming did reduce the level of soluble copper (Cu), zinc (Zn), aluminium (Al) and magnesium (Mg)...": The supplementary data supports the claim only in the case of Cu and Al
9. Line 273: Microbial communities naturally varied in their siderophore production (Imer: random intercept variance of microbial community = 0.15; Fig. 4a), with all ancestral communities containing multiple siderophore-producing isolates (Fig. 4c). In several cases the authors describe the data for Fig a followed by Fig c and then come back to Fig b. This is confusing. Either change the order of the figure panels or describe in order.
10. In Fig 4B controls show increased siderophore production. How do we explain this observation?

Decision letter (RSPB-2019-0804.R0)

13-May-2019

Dear Dr Hesse:

Your manuscript has now been peer reviewed and the reviews have been assessed by an Associate Editor. The reviewers' comments (not including confidential comments to the Editor) and the comments from the Associate Editor are included at the end of this email for your reference. As you will see, the reviewers and the Editors have raised some concerns with your manuscript and we would like to invite you to revise your manuscript to address them.

We do not allow multiple rounds of revision so we urge you to make every effort to fully address all of the comments at this stage. If deemed necessary by the Associate Editor, your manuscript will be sent back to one or more of the original reviewers for assessment. If the original reviewers are not available we may invite new reviewers. Please note that a revise decision is not a guarantee of eventual acceptance of your manuscript.

Research ethics:

Use of animals and field studies:

It is a condition of publication that you make available the data and research materials supporting the results in the article. Datasets should be deposited in an appropriate publicly available repository and details of the associated accession number, link or DOI to the datasets must be included in the Data Accessibility section of the article

(<https://royalsociety.org/journals/ethics-policies/data-sharing-mining/>). Reference(s) to datasets should also be included in the reference list of the article with DOIs (where available).

Please submit a copy of your revised paper within three weeks. If we do not hear from you within this time your manuscript will be rejected. If you are unable to meet this deadline please let us know as soon as possible, as we may be able to grant a short extension.

Best wishes,

Prof Sarah Brosnan
Editor, Proceedings B
mailto: proceedingsb@royalsociety.org

Associate Editor
Board Member: 1
Comments to Author:
Dear Dr Hesse

Your ms has been reviewed by two expert reviewers, and has split opinion. One reviewer recommends acceptance of the ms as it is, whereas the other is less positive in their assessment.

My view of reviewer 2's criticisms is that most of these points can be addressed by revising the ms to include additional explanation and caveats where appropriate.

Yours sincerely,
Mike

Reviewer(s)' Comments to Author:

Referee: 1

Comments to the Author(s)

In this report, the authors performed a simple microcosm experiment to assess the effect of anthropogenic disturbance (liming) on a number of microbial communities' ability to bioremediate pollution (heavy metal through siderophores). I really must commend the authors on an extremely well reasoned, designed and explained study. There is nothing I would amend to this study hence the recommendation to accept as is.

Referee: 2

Comments to the Author(s)

Manuscript: Anthropogenic remediation of heavy metals selects against natural microbial remediation by Hesse et al.

Dear authors,

Thank you for your manuscript "Anthropogenic remediation of heavy metals selects against natural microbial remediation". Here you study an interesting question whether artificial heavy metal detoxification strategies selects against microbial communities which perform natural remediation. The work has a great scope in various areas including microbial ecology, evolution and, health.

The observations from the study are: (a) liming increases pH of the soil (makes it slightly alkaline); (b) it decreases availability of metals (although only in 2 out of 11 metals studied); (c) iron was most abundant and importantly upon liming it's availability is not decreased; (d) after 12 weeks of liming treatment the microbial communities changed from mainly siderophore producers to non-producers. From these, authors conclude that anthropogenic strategies such as liming selects against natural detoxification.

The experimental design, sample size and the statistical analysis used by the authors are appropriate.

Here are my concerns:

(a) First of all, liming itself does not appear to be a great strategy for detoxification as it decreases the availability of only a few metals. This, however, is not the authors claim but I wonder if it would have been appropriate to study the effect of anthropogenic strategies which have shown to be efficient in detoxification.

(b) There is a correlation between increased pH (upon liming) and selection for siderophore non-producers. Siderophores are produced for chelating iron. Assuming siderophores also chelate other metals (to a large extent?) the mere selection of siderophore non-producers does support the conclusion that anthropogenic strategies selects against natural detoxification. Moreover, authors indirectly claim that natural remediation is efficient and artificial strategies affect this process. I guess the artificial strategies are used because natural process is either very slow or probably not efficient to detoxify the huge heavy metal load coming out from mining, etc.

Other concerns (as they appear in text):

1. Line 99: "To test whether liming selects against detoxification in natural microbial communities". I think this should be phrased as "To test whether liming selects against siderophore non-producer..." or something else like this. This is because detoxification is not under selection. It (if at all) is a consequence of selection on a particular bacterial 'phenotypes'.
2. Line 109: "after which the supernatant was stored..." Please mention if supernatant was treated in some way (filtered/centrifuged). This is because once vortexed the soil mixture probably was highly turbid and some floating particles.
3. Line 149: "Briefly, 5 μ l of DNA.." mention DNA concentration or total DNA (either ug or ug/ul).
4. Line 200: "mixture (CAA+CAS)." I wonder if this should have been CAA + water as the authors diluted assay solution with water.
5. Line 238: "(a) Liming reduces soil acidity and non-iron metal availability." Although in a relatively small number of samples but how do we explain increased pH in CONTROLS? Could this be because of natural properties of the bacterial communities in those samples? How do we explain the variation in red symbols? Here, the size of the symbol indicates change in pH which I thought should have been uniform. Besides, in some cases after liming the pH is decreased.
6. Line 246: "Iron was by far the most common metal across all samples and liming did not restrict its availability". Based on this result I wonder if the major conclusion of the MS can be interpreted differently. Authors claim that liming selects against natural detoxification. First of all, this should be liming selects siderophore non-producers. But, importantly, if iron availability is not affected and as authors point out the iron was abundant in their soil samples then one would expect siderophore non-producers to be favoured (probably because of the cost of basal level siderophore production).
7. Line 250: However, the availability of non-ferrous metals was significantly lower in lime-treated soils (..... $P = 0.047$). Authors probably will agree that the availability of non-ferrous metal appears to be statistically significant ($p = 0.047$) however these differences appear to be minor and probably does not mean much from the biology perspectives.
8. Line 253: "Notably, while none of the quantified metals was more readily available in lime-treated soils, liming did reduce the level of soluble copper (Cu), zinc (Zn), aluminium (Al) and magnesium (Mg)...": The supplementary data supports the claim only in the case of Cu and Al
9. Line 273: Microbial communities naturally varied in their siderophore production (Imer: random intercept variance of microbial community = 0.15; Fig. 4a), with all ancestral communities containing multiple siderophore-producing isolates (Fig. 4c). In several cases the authors describe the data for Fig a followed by Fig c and then come back to Fig b. This is confusing. Either change the order of the figure panels or describe in order.
10. In Fig 4B controls show increased siderophore production. How do we explain this observation?

Author's Response to Decision Letter for (RSPB-2019-0804.R0)

See Appendix A.

Decision letter (RSPB-2019-0804.R1)

28-May-2019

Dear Dr Hesse

I am pleased to inform you that your manuscript entitled "Anthropogenic remediation of heavy metals selects against natural microbial remediation" has been accepted for publication in Proceedings B. Thank you for such a nice revision that carefully addressed the concerns that were raised.

Open Access

You are invited to opt for Open Access, making your freely available to all as soon as it is ready for publication under a CC BY licence. Our article processing charge for Open Access is £1700. Corresponding authors from member institutions (<http://royalsocietypublishing.org/site/librarians/allmembers.xhtml>) receive a 25% discount to these charges. For more information please visit <http://royalsocietypublishing.org/open-access>.

Your article has been estimated as being 9 pages long. Our Production Office will be able to confirm the exact length at proof stage.

Paper charges

Sincerely,

Prof Sarah F. Brosnan
Editor, Proceedings B
mailto: proceedingsb@royalsociety.org

Associate Editor:
Board Member
Comments to Author:
(There are no comments.)

Appendix A

Dear Editor,

We would like to thank the referees for their critical and useful comments on our manuscript: “Anthropogenic remediation of heavy metals selects against natural microbial remediation” (RSPB-2019-0804). Please find below our detailed point-by-point response.

Yours sincerely,

Elze Hesse, Daniel Padfield, Florian Bayer, Eleanor M. van Veen, Christopher G. Bryan & Angus Buckling

Referee: 1

In this report, the authors performed a simple microcosm experiment to assess the effect of anthropogenic disturbance (liming) on a number of microbial communities' ability to bioremediate pollution (heavy metal through siderophores). I really must commend the authors on an extremely well reasoned, designed and explained study. There is nothing I would amend to this study hence the recommendation to accept as is.

We are very grateful the reviewer appreciated our work.

Referee: 2

Dear authors, Thank you for your manuscript “Anthropogenic remediation of heavy metals selects against natural microbial remediation”. Here you study an interesting question whether artificial heavy metal detoxification strategies selects against microbial communities which perform natural remediation. The work has a great scope in various areas including microbial ecology, evolution and, health.

The observations from the study are: (a) liming increases pH of the soil (makes it slightly alkaline); (b) it decreases availability of metals (although only in 2 out of 11 metals studied); (c) iron was most abundant and importantly upon liming its availability is not decreased; (d) after 12 weeks of liming treatment the microbial communities changed from mainly siderophore producers to non-producers. From these, authors conclude that anthropogenic strategies such as liming selects against natural detoxification.

The experimental design, sample size and the statistical analysis used by the authors are appropriate.

Here are my concerns:

(a) First of all, liming itself does not appear to be a great strategy for detoxification as it decreases the availability of only a few metals. This, however, is not the authors' claim but I wonder if it would have been appropriate to study the effect of anthropogenic strategies that have shown to be efficient in detoxification.

The main reason we opted for liming is that it is a common economical method for remediating acidic metal-polluted soils. We acknowledge the reviewer's concern, and agree it would be worthwhile to study additional (phyto)-remediation strategies in the future. However, we would also like to point out that while liming did not immobilize heavy metals to a similar extent, it did significantly reduce total non-iron metal availability.

(b) There is a correlation between increased pH (upon liming) and selection for

siderophore non-producers. Siderophores are produced for chelating iron. Assuming siderophores also chelate other metals (to a large extent?) the mere selection of siderophore non-producers does support the conclusion that anthropogenic strategies selects against natural detoxification. Moreover, authors indirectly claim that natural remediation is efficient and artificial strategies affect this process. I guess the artificial strategies are used because natural process is either very slow or probably not efficient to detoxify the huge heavy metal load coming out from mining, etc.

The reviewer is correct in their assertions that liming is beneficial in heavily contaminated soils, and we clarify this point in the introduction: “Hence, lime-containing materials are commonly applied to severely metal-contaminated soils...”. Please note that the idea that siderophores chelate metals other than iron is not merely an assumption, but has been experimentally demonstrated (studies cited in the Introduction).

Other concerns (as they appear in text):

1. Line 99: “To test whether liming selects against detoxification in natural microbial communities”. I think this should be phrased as “To test whether liming selects against siderophore non-producer...” or something else like this. This is because detoxification is not under selection. It (if at all) is a consequence of selection on a particular bacterial ‘phenotypes’.

We have changes this accordingly: “To test whether liming selects against siderophore production in natural microbial communities”.

2. Line 109: “after which the supernatant was stored...” Please mention if supernatant was treated in some way (filtered/centrifuged). This is because once vortexed the soil mixture probably was highly turbid and some floating particles.

We did not filter soil wash supernatants, because these were not directly used in our CAS assays (see section e Methods); hence sample turbidity was not relevant. Instead, we used plated serial-diluted soil washes to pick twenty-four individual isolates per community to obtain measures of siderophore production. We have clarified this by removing supernatant from the sentence: “Freezer stocks were prepared by vortexing 1 g of soil for 1 min with 6 ml of M9 buffer and sterile glass beads, after which the soil washes were stored at -80°C in a final concentration of 25% glycerol”.

3. Line 149: “Briefly, 5 µl of DNA..” mention DNA concentration or total DNA (either ug or ug/ul).

We have now included this information: “Briefly, 5 µl of DNA (mean ± SD concentration = 15.99 ± 11.80 ng / µl) entered a first round of PCR with cycle....”

4. Line 200: “mixture (CAA+CAS).” I wonder if this should have been CAA + water as the authors diluted assay solution with water.

Instead of adding water to individual supernatants, we diluted the actual CAS solution. We have clarified this: “..using the liquid CAS assay described by Schwyn and Neilands [44], modified such that one volume of ddH₂O was added to the CAS solution [45]. Siderophore production per isolate was estimated using: $[1 - (A_i/A_{ref})] / [OD_i]$, where OD_i = optical density at 600 nm and A_i = absorbance at 630 nm of the assay mixture (supernatant + CAS solution) and A_{ref} = absorbance at 630 nm of reference mixture (CAA + CAS solution)”.

5. Line 238: “(a) Liming reduces soil acidity and non-iron metal availability.” Although in a relatively small number of samples, but how do we explain increased pH in CONTROLS? Could this be because of natural properties of the bacterial communities in those samples? How do we explain the variation in red symbols? Here, the size of the symbol indicates change in pH, which I thought should have been uniform. Besides, in some cases after liming the pH is decreased.

An increase in pH in the control microcosms could result from changes in microbial traits as well as from changes in abiotic factors resulting from laboratory incubation. We clarify this point in the results: “The increase in siderophore production in control soils through time (Fig 4b) could be driven by a range of abiotic and biotic changes associated with growth under laboratory conditions”. We are hesitant to speculate on the exact mechanism underlying this observation, as these could be manifold. Crucially, this lack of knowledge does not alter the interpretation of our main finding of human intervention selecting against metal-detoxifying siderophores.

While the effect of liming likely varies across soils with different geochemical properties, liming is expected to have a greater neutralizing effect on soils with lower pH (e.g. difference between pH 3-7 > pH 5-7) and this is exactly what we find.

6. Line 246: “Iron was by far the most common metal across all samples and liming did not restrict its availability”. Based on this result I wonder if the major conclusion of the MS can be interpreted differently. Authors claim that liming selects against natural detoxification. First of all, this should be liming selects siderophore non-producers. But, importantly, if iron availability is not affected and as authors point out the iron was abundant in their soil samples then one would expect siderophore non-producers to be favoured (probably because of the cost of basal level siderophore production).

If iron availability did indeed impose strong selection on siderophore production (by favouring non-producers as suggested), then this response should be similar across treatments (as iron concentrations were similar). In contrast, liming strongly selected against siderophore production, which was maintained in the paired control microcosms. This would strongly suggest that selection against siderophores resulted from lime addition reducing total availability of non-ferrous metals. This point was made in the first paragraph of the discussion, but we spell this out even more explicitly: “In other words, selection imposed on siderophore production as a consequence of iron availability would have differed little between limed and control soils”.

7. Line 250: However, the availability of non-ferrous metals was significantly lower in lime-treated soils (..... $P = 0.047$). Authors probably will agree that the availability of non-ferrous metal appears to be statistically significant ($p = 0.047$) however these differences appear to be minor and probably does not mean much from the biology perspectives.

We kindly disagree with the reviewer on this particular item – although changes in non-ferrous metal concentrations indeed seemed to be minor, they did cause a massive shift in siderophore production. This would strongly suggest that even minor changes in non-ferrous metal availability are of biological relevance to microbial communities inhabiting metal-contaminated soils.

8. Line 253: “Notably, while none of the quantified metals was more readily available in lime-treated soils, liming did reduce the level of soluble copper (Cu), zinc (Zn), aluminium

(Al) and magnesium (Mg)...”: The supplementary data supports the claim only in the case of Cu and Al

We agree with the reviewer and have clarified this in the text and accompanying figure 2.

9. Line 273: Microbial communities naturally varied in their siderophore production (lmer: random intercept variance of microbial community = 0.15; Fig. 4a), with all ancestral communities containing multiple siderophore-producing isolates (Fig. 4c). In several cases the authors describe the data for Fig a followed by Fig c and then come back to Fig b. This is confusing. Either change the order of the figure panels or describe in order.

We have changed the order accordingly.

10. In Fig 4B controls show increased siderophore production. How do we explain this observation?

Please see point 5.